# Genetic and Evolutionary Analysis of Porcine Deltacoronavirus in Guangxi Province, Southern China, from 2020 to 2023

**DOI:** 10.3390/microorganisms12020416

**Published:** 2024-02-19

**Authors:** Biao Li, Yeheng Gao, Yan Ma, Kaichuang Shi, Yuwen Shi, Shuping Feng, Yanwen Yin, Feng Long, Wenchao Sun

**Affiliations:** 1College of Animal Science and Technology, Guangxi University, Nanning 530005, China; libiao6096@126.com (B.L.); yanmayy@163.com (Y.M.); shiyuwen2@126.com (Y.S.); 2Institute of Agricultural and Animal Husbandry Industry Development, Guangxi University, Nanning 530005, China; gyh1253814494@163.com; 3Guangxi Center for Animal Disease Control and Prevention, Nanning 530001, China; fsp166@163.com (S.F.); yanwen0349@126.com (Y.Y.); longfeng1136@163.com (F.L.); 4Institute of Virology, Wenzhou University, Wenzhou 325035, China

**Keywords:** porcine deltacoronavirus, S gene, M gene, N gene, phylogenetic analysis, recombination

## Abstract

Porcine deltacoronavirus (PDCoV) has shown large-scale global spread since its discovery in Hong Kong in 2012. In this study, a total of 4897 diarrheal fecal samples were collected from the Guangxi province of China from 2020 to 2023 and tested using RT-qPCR. In total, 362 (362/4897, 7.39%) of samples were positive for PDCoV. The S, M, and N gene sequences were obtained from 34 positive samples after amplification and sequencing. These PDCoV gene sequences, together with other PDCoV S gene reference sequences from China and other countries, were analyzed. Phylogenetic analysis revealed that the Chinese PDCoV strains have diverged in recent years. Bayesian analysis revealed that the new China 1.3 lineage began to diverge in 2012. Comparing the amino acids of the China 1.3 lineage with those of other lineages, the China 1.3 lineage showed variations of mutations, deletions, and insertions, and some variations demonstrated the same as or similar to those of the China 1.2 lineage. In addition, recombination analysis revealed interlineage recombination in CHGX-MT505459-2019 and CHGX-MT505449-2017 strains from Guangxi province. In summary, the results provide new information on the prevalence and evolution of PDCoV in Guangxi province in southern China, which will facilitate better comprehension and prevention of PDCoV.

## 1. Introduction

Porcine deltacoronavirus (PDCoV) causes a gastrointestinal infectious disease characterized by fever, vomiting, watery diarrhea, dehydration, and loss of appetite in pigs of different ages, especially piglets [1]. The autopsy of piglets that die of the disease shows pathological changes such as blunting and atrophic changes of the intestinal villi and necrosis of the villous cells [2]. These symptoms are similar to those of other viruses that cause gastrointestinal manifestations in pigs, and it is difficult to distinguish them only based on the clinical signs and gross pathological changes [3,4].

PDCoV, which belongs to the genus *Deltacoronavirus* in the family *Coronaviridae* of order *Nidovirales*, is a single-stranded, positive-sense RNA virus [5]. The virus was first identified and named HKU15 coronavirus in Hong Kong, China, in 2012 [6], but it was first confirmed to be the pathogenic virus of porcine diarrheal disease in the United States in 2014. Since then, PDCoV has been confirmed in many countries such as Thailand [7], Vietnam [8], Japan [9], Korea [10], Canada [11], and Mexico [12], which seriously endangered the health of pigs worldwide [13]. Notably, the blood from three children in Haiti’s schools during 2014–2015 were discovered to be positive for PDCoV, which was the first report on human infection with PDCoV [14], indicating the zoonotic potential of PDCoV.

PDCoV has a genome of about 25.4 kb in length, with an arrangement of 5′ untranslated region (UTR), open reading frame 1a/1b (ORF1a/1b), spike (S), envelope (E), membrane (M), nonstructural protein 6 (NS6), nucleocapsid (N), NS7, NS7a, and 3′ UTR [1]; of the encoded proteins, S, M, and N proteins play important roles. The PDCoV S protein can induce neutralizing antibodies [15,16]. In addition, the infection of PDCoV in human cells is associated with the ability of the S protein to recognize and bind to cellular receptors [17,18]. The N protein is a viral capsid’s component and is considered to be the most abundant and multifunctional protein, which performs various biological functions during viral replication and translation [19,20]. The M protein is critical in the assembly of the virion and virus–host interactions [21,22]. In order to learn more about the epidemiological characteristics and genetic properties of PDCoV in recent years, the S, M, and N genes of PDCoV from the Guangxi province in southern China were selected for amplification, sequencing, and analysis.

## 2. Materials and Methods

### 2.1. Collection and Detection of Specimens for PDCoV

From October 2020 to October 2023, a total of 4897 diarrheal fecal specimens were collected from different pig farms in different regions of Guangxi province in southern China. The specimens were transported to our laboratory under ≤4 °C conditions within 12 h after collection and immediately processed to extract total RNA or stored at −80 °C before being used.

The specimens were resuspended using pH7.2 phosphate-buffered saline (PBS) (in a ratio of 1:4, w/v), vortexed (5 min), and centrifuged at 4 °C (12,000 rpm, 10 min). The total RNA was extracted from supernatants using MiniBEST Viral RNA/DNA Extraction Kit Ver.5.0 (TaKaRa, Dalian, China; Cat No. 9766), then immediately used to test PDCoV or stored at −80 °C until used.

The multiplex real-time quantitative RT-PCR (RT-qPCR) using the Q5 qPCR system (ABI, Carlsbad, CA, USA) for detection of PDCoV, transmissible gastroenteritis virus (TGEV), porcine epidemic diarrhea virus (PEDV), and swine acute diarrhea syndrome coronavirus (SADS-CoV) established in our laboratory [23] were used to test all the samples. The specific primers and probes used for detection of PDCoV, TGEV, PEDV, and SADS-CoV were as follows: PDCoV(M)-U: ATCGACCACATGGCTCCAA, PDCoV(M)-D: CAGCTCTTGCCCATGTAGCTT, PDCoV(M)-P: FAM-CACACCAGTCGTTAAGCATGGCAAGCT-BHQ1; TGEV(M)-U: GCAATTCTTTGCGTTAGTGCAT, TGEV(M)-D: AGCGTACAAATTCCCTGAAAGC, TGEV(M)-P: Texas Red-CTTCCTCTCGAAGGTGTGCCAACTGG-BHQ2; PEDV(N)-U: CTGGAATGAGCAAATTCGCTG, PEDV(N)-D: CAACCCAGAAAACACCCTCAG, PEDV(N)-P: JOE-AGCGAATTGAACAACCTTCCAATTGGCA-BHQ1; SADS-CoV(N)-U: TACTGGTCCTCACGCAGATG, SADS-CoV(N)-D: ACGATTGCGAACACCAAGAC; SADS-CoV(N)-P: Cy5-CAACAGCGACCCAATGCACACCCT-BHQ3. The recombinant standard plasmid construct containing PDCoV M gene fragment was previously prepared for PDCoV as described by Zhou et al. [23], and used as positive control. The negative fecal and nuclease-free sterilized distilled water were used as negative controls. The reaction system and the procedure of the multiplex RT-qPCR were described by Zhou et al. [23]. In brief, the reaction system contained One Step PrimeScript™ RT-PCR Kit (Perfect Real Time) (TaKaRa, Dalian, China; Cat No. RR064A) which included One-Step RT-PCR Buffer III, Ex Taq HS, and PrimeScript RT Enzyme Mix II (RNA/DNA), primers/probes PDCoV(M)-U/D/P, TGEV(M)-U/D/P, PEDV(N)-U/D/P, and SADS-CoV(N)-U/D/P, nucleic templates, and nuclease-free distilled water to a final volume of 20 µL. The procedures were as follows: 42 °C 5 min, 95 °C 10 s, and then 40 cycles of 95 °C 5 s, 57 °C 34 s. The fluorescence signals were automatically recorded at the end of each cycle.

Finally, 362 specimens were tested positive for PDCoV using the multiplex RT-qPCR [23], which were distributed in different regions of Guangxi province (Figure 1). They were further used to finish the following experiments.

### 2.2. Amplification and Sequencing

Of the 362 PDCoV-positive specimens using multiplex RT-qPCR [23], 34 specimens were selected based on different regions and different pig farms in Guangxi province, different seasons of the year, and having Ct values less than 25 cycles. They were used to amplify the S (3480 bp), M (654 bp), and N (1029 bp) genes of PDCoV by RT-PCR using the amplification primers (Table 1). The total nucleic acids were extracted using MiniBEST Viral RNA/DNA Extraction Kit Ver.5.0 (TaKaRa, Dalian, China; Cat No. 9766) and reverse transcribed to cDNA using PrimeScript II 1st Strand cDNA Synthesis Kit (TaKaRa, Dalian, China; Cat No. 6210A). The amplification system (50 μL): 2 × Taq PCR Master Mix (TaKaRa, Dalian, China; Cat No. RR901A) 25 μL, forward and reward primers (20 pmol/μL) 0.8 μL each, sterilized distilled water 18.4 μL, cDNA 5 μL. The amplification procedure: 95 °C 5 min; 35 cycles of 95 °C 30 s, 58.6 °C 30 s, and 72 °C for 60 s.

The PCR products were purified using MiniBEST DNA Fragment Purification Kit Ver.4.0 (TaKaRa, Dalian, China; Cat No. 9761) after 1% agarose gel electrophoresis, ligated into pMD18-T vector (TaKaRa, Dalian, China; Cat No. 6011), transformed into *DH5α* competent cells (TaKaRa, Dalian, China; Cat No. 9057). The positive clones were cultured at 37 °C for 20–24 h, and the plasmids were extracted using MiniBEST Plasmid Purification Kit Ver.4.0 (TaKaRa, Dalian, China; Cat No. 9760) and sent to IGE biotechnology LTD (Guangzhou, China) for sequencing. After splicing different gene fragment sequences, the complete S and N gene sequences were obtained. The obtained gene sequences were validated by BLAST analysis at the National Center for Biotechnology Information (NCBI) “https://blast.ncbi.nlm.nih.gov/Blast.cgi (accessed on 18 December 2023)”. Finally, a total of 34 PDCoV S, M, and N gene sequences were obtained, respectively (Appendix A).

### 2.3. Sequence Comparison and Phylogenetic Analysis

The S, M, and N gene sequences of PDCoV from Guangxi province published in NCBI “https://www.ncbi.nlm.nih.gov/nucleotide/ (accessed on 18 December 2023)” were downloaded. Finally, 37 S gene sequences, 32 M gene sequences, and 34 N gene sequences were collected from NCBI (Appendix A). The other S, M, and N gene sequences from other provinces in China and from other countries such as the United States, Republic of Korea, Vietnam, Laos, Japan, and Thailand, were also collected from NCBI. The obtained gene sequences in this study, together with all the collected sequences from NCBI (Appendix A), i.e., 217 S, 200 M, and 202 N gene sequences, were analyzed using the Clustal W algorithm of DNAstar 7.0 software “https://www.dnastar.com/software/ (accessed on 18 December 2023)”. The phylogenetic trees were constructed using the maximum likelihood method of MEGA Ⅹ software “https://www.megasoftware.net/archived_version_active_download (accessed on 18 December 2023)”, which was optimized through the online website Interactive Tree of Life (iTOL) “https://itol.embl.de/ (accessed on 18 December 2023)” [24].

### 2.4. Bayesian Temporal Dynamics Analysis

The 71 PDCoV S gene sequences from Guangxi province were compared using MEGA Ⅹ software, the maximum likelihood tree was reconstructed using IQ-TREE software “http://iqtree.cibiv.univie.ac.at/ (accessed on 18 December 2023)”, the TN+F+R3 model was implemented by 1000 bootstrap. In order to determine the temporal structure, it was verified by regression of root-tip genetic distance of the sequences using TempEst software [25,26]. Bayesian Markov chain Monte Carlo (MCMC) was chosen to infer the dispersion time of the PDCoV S gene in BEAST v1.10.4 “http://beast.community/ (accessed on 18 December 2023)”, and a suitable replacement model was calculated through the ModelFinder option of the PhyloSuite software [27]. For the S gene, the final choice was used with the relaxed molecular clock, TN93+F+G4 replacement model, and Bayesian skyline model replacement and after which the MCMC was run in parallel on the 3 chains for 200 million steps with 10% aging. The post-run data was visualized through Tracer software, and only data with all parameters ESS > 200 was considered to be valid. The data was calculated through TreeAnnotator v1.10.4. The MCMC tree was annotated after aging by 10% and visualized using FigTree version 1.4.4 “http://beast.community/ (accessed on 18 December 2023)”.

### 2.5. Genome Recombination Events Analysis

Recombination analysis was performed on all S gene sequences using the Recombination Detection Program (RDP4) software, which was downloaded from the public website “http://www.bioinf.manchester.ac.uk/recombination/programs (accessed on 18 December 2023)” with a window size of 500 bp and a *p*-value of 0.05. Seven algorithms were selected for recombination analysis according to the recommendations of the RDP manual, including RDP, GENECONY, BootScan, MaxChi, Chimaera, SiScan, and 3Seq in RDP4 software. Sequences were considered to be potentially recombinant only if at least six algorithms supported them; for the potentially recombinant sequences, it was verified using the SimPlot software “https://github.com/Stephane-S/Simplot_PlusPlus (accessed on 18 December 2023)”.

## 3. Results

### 3.1. Detection Results of the Clinical Specimens

The 4897 clinical specimens were tested using multiplex RT-qPCR [23], and 362 specimens (7.39%, 362/4897) were positive for PDCoV. The positive rates of the clinical specimens were 14.85% (30/202), 9.70% (86/887), 8.60% (232/2698), and 1.26% (14/1110) from 2020 to 2023, respectively. The positive specimens were distributed in different regions in Guangxi province (Figure 1). A total of 34 positive specimens were selected to amplify and sequence the S, M, and N genes. Finally, 34 S, 34 M, and 34 N gene sequences were obtained and were uploaded to the NCBI GenBank “https://www.ncbi.nlm.nih.gov/genbank/ (accessed on 18 December 2023)” under the accession numbers: OR659117-OR659150 for S gene, OR659151-OR659184 for M gene, and OR659185-OR659218 for N gene.

### 3.2. Phylogenetic Analysis Basing on S Gene Sequences

To understand the phylogenetic information of the S gene, a phylogenetic tree was generated based on the information of a total of 217 gene sequences (Appendix A), including 34 S gene sequences obtained in this study, 37 S gene sequences from Guangxi province and 146 S gene sequences from other Chinese provinces and other countries downloaded from GenBank in NCBI (Figure 2). Since there is no standard method for systematically subgrouping PDCoV at present, the subgroups of the genetic evolutionary tree were categorized in a previous report [28]. All the sequences could be classified into different subgroups relating to the source of different countries/regions, i.e., the prototype (early China) lineage, the USA lineage, the Southeast Asia lineage, the China 1.1 lineage, the China 1.2 lineage, and the China 1.3 lineage. The homology of the S gene in each lineage in China is shown in Table 2. Most sequences from Guangxi province were distributed in the subgroups from China, but some other sequences were distributed in the subgroup from Southeast Asia. In particular, the 34 S gene sequences obtained in this study, together with the sequences from other Chinese provinces such as Hunan, Shandong, Hong Kong, and other provinces in recent years, were located in a newly discovered lineage in this study, i.e., the China 1.3 lineage.

### 3.3. Phylogenetic Analysis Basing on M Gene Sequences

The phylogenetic tree was generated based on 200 M gene sequences (Appendix A), including 34 sequences obtained in this study, 32 sequences from Guangxi province, and 134 reference sequences from other Chinese provinces and other countries downloaded from GenBank in NCBI (Figure 3). All the M gene sequences were distributed into the Chinese branch, the American branch, and the Southeast Asian branch, of which the Chinese sequences had a wide distribution. All the M gene sequences obtained in this study were located in the Chinese subgroups.

### 3.4. Phylogenetic Analysis Basing on N Gene Sequences

The phylogenetic tree was generated based on 202 N gene sequences (Appendix A), including 34 sequences obtained in this study, 34 sequences from Guangxi province, and 134 reference sequences from other Chinese provinces and other countries downloaded from GenBank in NCBI (Figure 4). Three Vietnamese strains were located in the Chinese subgroups and were closer to the sequences from Guangxi province, including those obtained in this study.

### 3.5. Bayesian Temporal Dynamics Analysis

The phylogenetic tree was consistent with the constructed MCC tree, which is based on the PDCoV S gene with the PDCoV temporal scale (Figure 5). The MCC tree indicated that all PDCoV strains were divided into five genotypes, i.e., China 1.1, China 1.2, China 1.3, USA, and Southeast Asia branches (Figure 5a), and the branch of China 1.3 lineage had a later differentiation time than the others, and may be a newly appearing branch in recent years. The Bayesian skyline (Figure 5b) showed the practical population size map of PDCoV transmission in Guangxi province in recent years. After PDCoV was discovered in Hong Kong in 2012, the large-scale transmission of this virus also occurred in Guangxi province, peaked in 2015, then weakened until 2020, and kept a stable trend onwards. The fluctuation of the adequate population size of the Bayesian skyline was similar to a previous report [29].

### 3.6. Recombination Analysis of S Gene

The RDP4 software was used for recombination event analysis of the S gene (Figure 6), and two strains from Guangxi province showed recombination signals. The CHGX-MT505459-2019 strain originated from the recombination that occurred between the CHGX-MT505446-2017 and CHGX-MT505452-2018 strains. Among them, the CHGX-MT505446-2017 strain was the primary parent with 98.9% similarity, and the CHGX-MT505452-2018 strain was the minor parent with 99.4% similarity, and the potential breakpoints were found in the region of 586 nt–1390 nt. The other strain, CHGX-MT505449-2017, was derived from the CHGX-MN173782-2016 strain [30] and a recombination event between the CHGX-MT505447-2017 strain, with the CHGX-MN173782-2016 strain as the primary parent with 99.5% similarity and the CHGX-MT505447-2017 strain as the secondary parent with 100% similarity, and the potential breakpoints were found in the region of 1375 nt–2055 nt.

### 3.7. Genetic Evolution Rates of S, M, and N Genes

The genetic evolution rate of PDCoV S, M, and N genes in Guangxi province was analyzed using BEAST software “http://beast.community/ (accessed on 18 December 2023)” (Table 3). The S, M, and N genes had evolutionary rates of 1.907 × 10^−3^, 8.321 × 10^−4^, and 1.135 × 10^−3^ substitutions/site/year, respectively, indicating that the S gene had the highest evolution rate.

### 3.8. Deduence Analysis of the S Protein

To know more about the genetic characteristics of the China 1.3 lineage, all the amino acid sequences of this branch and some amino acid sequences of other branches were put into BioEdit software “https://www.bioedit.com/ (accessed on 18 December 2023)” for comparison and analysis. The sequence of the earliest PDCoV strain CHAN-KP757890-2004 (GenBank accession no. KP757890), was used as the primary reference strain (Figure 7).

The sequences of China 1.3 lineage were compared with other branches, and the mutations, deletions, and insertions were found in this branch. Interestingly, many mutations in this branch are identical or similar to those in the China 1.2 lineage, including mutations at positions S28T and Y31H/N, and deletion at position 44, and these variations were consistent in all the strains in the China 1.3 lineage. In addition, the strains showed the mutations at positions D114N, S166A, T726N, and R786K, the insertions at position T178, and the deletions at positions 165, 180, 410, 465, 488, 489, 618, and 643, and these variations were same as or similar to those in the China 1.2 lineage (Figure 7).

## 4. Discussion

PDCoV was first discovered in Hong Kong in 2012 and has been reported in many countries around the world [7,8,9,10,11,12,13]. PDCoV can infect pigs of all ages, especially piglets [1,5,6]. In recent years, the prevalent situations and evolution characterizations of PDCoV in different provinces in China have been reported [31,32,33,34], but those of Guangxi province have not been studied in detail until now. Guangxi raises approximately 60 million pigs annually, ranking as the sixth among the 34 provinces in China. It is necessary to understand the genetic and evolutionary characteristics of PDCoV in Guangxi province.

A total of 4897 fecal specimens from Guangxi province from October 2020 to October 2023 were tested using multiplex RT-qPCR [23]. A total of 362 specimens (7.39%, 362/4897) were positive for PDCoV, and the positivity rates decreased from 14.85% in 2020 to 1.26% in 2023. This indicated that the positivity rates of PDCoV in the clinical specimens decreased gradually after 2020, which may be attributed to strict biosecurity and effective measures performed in different pig farms since the outbreak of African swine fever (ASF) in China in 2018 [35,36], even if there no legal vaccine on the Chinese market for PDCoV until now.

Since S [15,16,37], M [21,22], and N [19,20] genes play important roles in viral replication, pathogenicity, pathogenesis, and immune response, S, M, and N genes are usually used as the targeted genes to perform a study on molecular epidemiology. PDCoV strains from different countries are usually categorized into four lineages based on the S gene sequence, including China, Early China, the United States, and Southeast Asia lineages [28,31,38,39], indicating the high genetic diversity of circulating strains of PDCoV. It has been reported that the geographic transmission patterns of the lineages in different subgroups are different, and more events of intralineage and interlineage recombination leading to higher viral genetic diversity have been found in the Chinese lineages [31]. In this study, the S, M, and N gene sequences obtained in Guangxi province during 2020–2023 showed 97.1–99.6%, 97.7–100%, and 96.1–100% homology to the reference sequences, respectively, indicating variations in the S gene were higher than those in the M and N genes. In addition, the genetic evolution rate of the S gene (1.907 × 10^−3^ substitutions/site/year) is faster than those of M (8.321 × 10^−4^ substitutions/site/year), and N (1.135 × 10^−3^ substitutions/site/year) genes. These genetic evolution rates were similar to the results of the previous reports [28,31].

The genetic evolution tree of the S gene revealed a new branch in the Chinese genealogy, namely, the China 1.3 lineage. Some mutations, deletions, and insertions in the China 1.3 lineage were similar to those in the China 1.2 lineage. It is noteworthy that some of the sequences from Guangxi province were not distributed in the same branch as those from other countries. The M and N genes are relatively conserved. The genetic evolution tree based on M and N genes revealed that some sequences in Guangxi province were distributed in the same branch with some Vietnamese strains in the Southeast Asian lineage. This may be related to the geographic location of Guangxi province, which is located in southern China and adjacent to Vietnam, and acted as an essential hub for exchanges between China and Southeast Asian countries. The legal and illegal personnel exchanges and goods trade between two countries are extremely frequent, which might help the virus to spread between these countries.

The Bayesian analysis indicated that the China 1.3 lineage diverged later and might be a newly emerged branch. Through the Bayesian skyline analysis, the PDCoV in Guangxi province has also appeared to be massively spread since it was discovered in Hong Kong in 2012, reached a peak in 2015, then showed a steady downward trend, and was followed by a slight increase in 2020, showing a steady trend in recent years. These results had similarities with the results of the Bayesian skyline performance mapped in a previous report for Asian PDCoV strains [29].

Recombination and variation usually contribute to the viral high genetic diversity [31,40]. A typical pattern of genetic evolution in coronaviruses is the occurrence of interspecies recombination [41]. Recombinants between different PDCoV strains have been reported in previous reports [31,42,43]. Therefore, the recombination events of all the S gene sequences from Guangxi province were analyzed in this study, and two strains with recombination signals were found. The CHGX-MT505459-2019 strain and the CHGX-MT505449-2017 strain, which were also previously obtained in our laboratory. Furthermore, no recombination signal was found from the sequences obtained from Guangxi province in this study. In addition, mutation, deletion, or insertion in the coronavirus S gene increases the genetic diversity and even changes the viral virulence [44,45]. The PDCoV S gene sequences from Guangxi province, which were distributed in the China 1.3 lineage, indicated the same or similar variation to those in the China 1.2 lineage. Most strains in the China 1.2 lineage came from Sichuan [45] and Shandong [33] provinces, while most strains from other provinces in China were distributed in the China 1.1 lineage according to the genetic evolution tree. The high genetic diversity of the circulating PDCoV strains is one of the key factors that need to be considered for effective prevention and control of this disease.

## 5. Conclusions

PDCoV showed a positivity rate of 7.39% in the clinical specimens during 2020–2023 in Guangxi province of southern China. A new branch, i.e., the China 1.3 lineage, was found according to the phylogenetic analysis based on S gene sequences and showed unique and consistent mutations at positions S28T, Y31H/N, and deletion at position 44 in all the strains. The recombination events were found in the circulating PDCoV strains in Guangxi province. These results provide more information on the prevalence and evolution of PDCoV in Guangxi province in southern China, which will facilitate better comprehension and prevention of PDCoV.

## Figures and Tables

**Figure 1 microorganisms-12-00416-f001:**
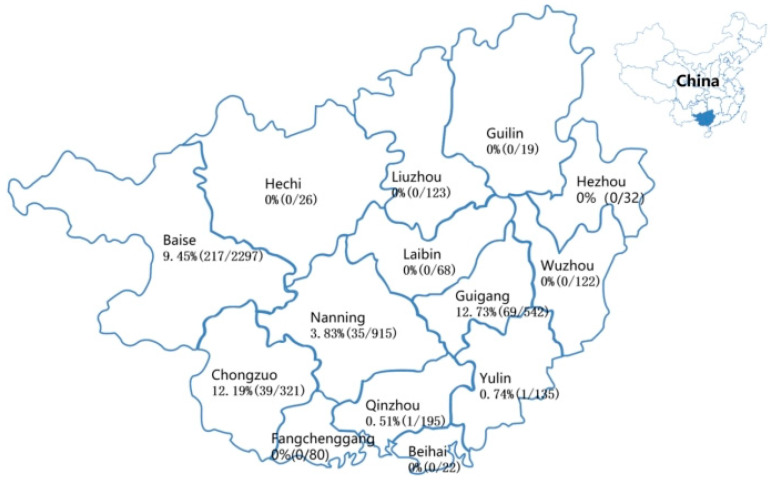
PDCoV distribution in Guangxi province. The positivity rates of PDCoV are marked for different regions.

**Figure 2 microorganisms-12-00416-f002:**
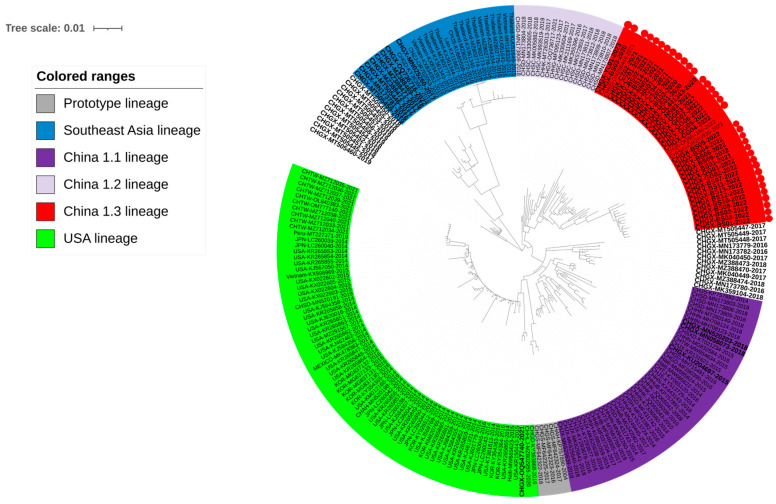
The phylogenetic tree based on PDCoV S gene nucleotide sequences. The TN93+G+I model is selected using the MEGA Ⅹ. The sequences from Guangxi province are bolded, and the sequences obtained in this study are marked with red circles.

**Figure 3 microorganisms-12-00416-f003:**
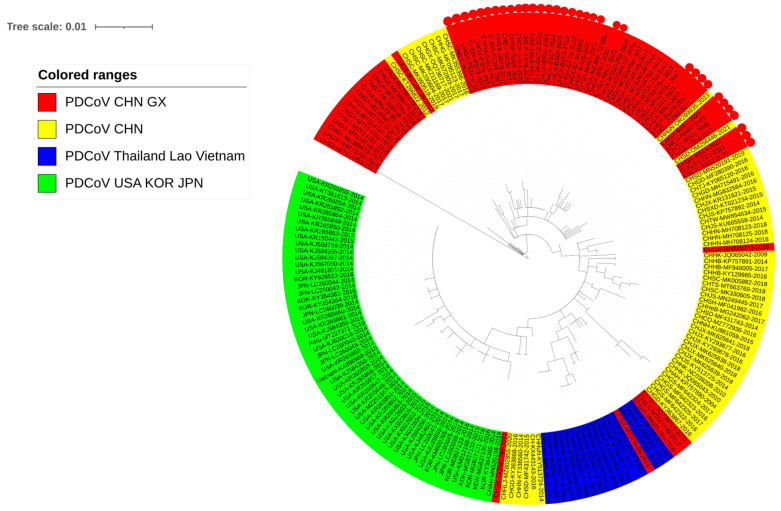
The phylogenetic tree based on PDCoV M gene nucleotide sequences. The K2+G model is selected using MEGA Ⅹ. The sequences from Guangxi province are bolded, and the sequences obtained in this study are marked with red circles.

**Figure 4 microorganisms-12-00416-f004:**
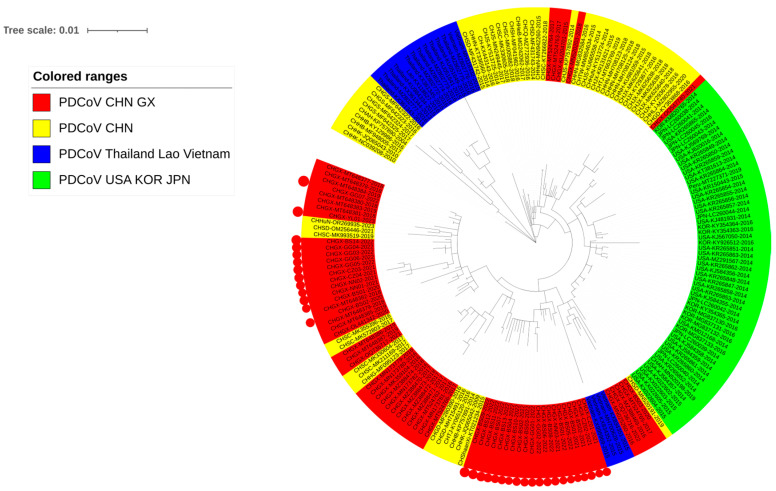
The phylogenetic tree based on PDCoV N gene nucleotide sequences. The K2+G+I model is selected using MEGA Ⅹ. The sequences from Guangxi province are bolded, and the sequences obtained in this study are marked with red circles.

**Figure 5 microorganisms-12-00416-f005:**
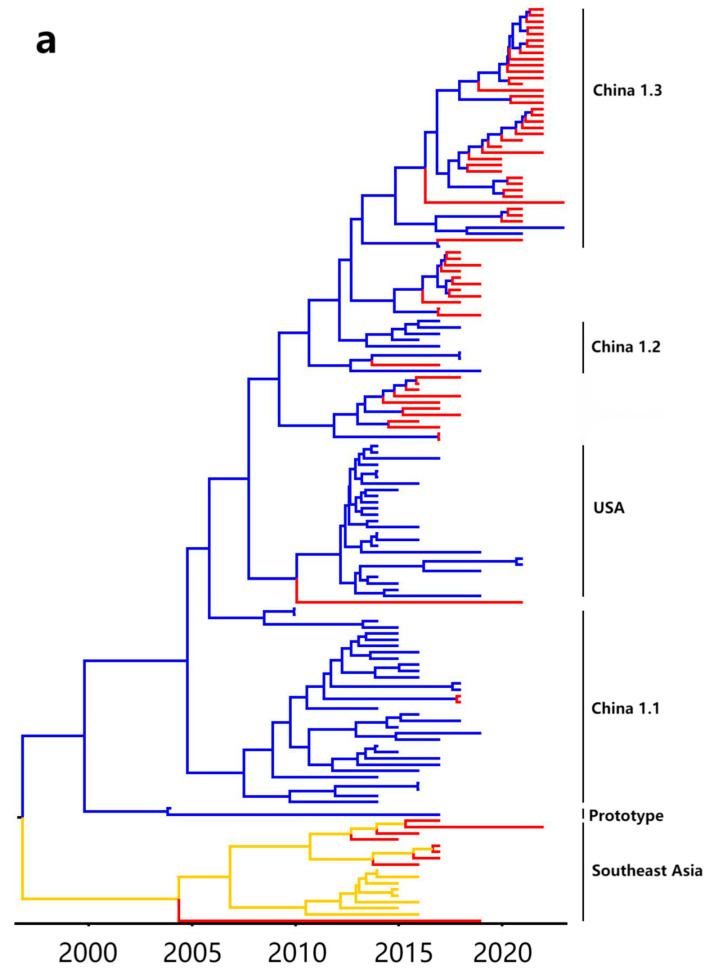
(**a**) The maximum clade credibility (MCC) tree generated by analyzing PDCoV S gene nucleotide sequences. Two subgroups are labeled using blue and orange, and the sequences obtained in Guangxi province are labeled using red. (**b**) Bayesian skyline of PDCoV S gene sequences in Guangxi province. The dark blue line indicates the average of the genetic diversity, and the light blue shading indicates a 95% confidence interval.

**Figure 6 microorganisms-12-00416-f006:**
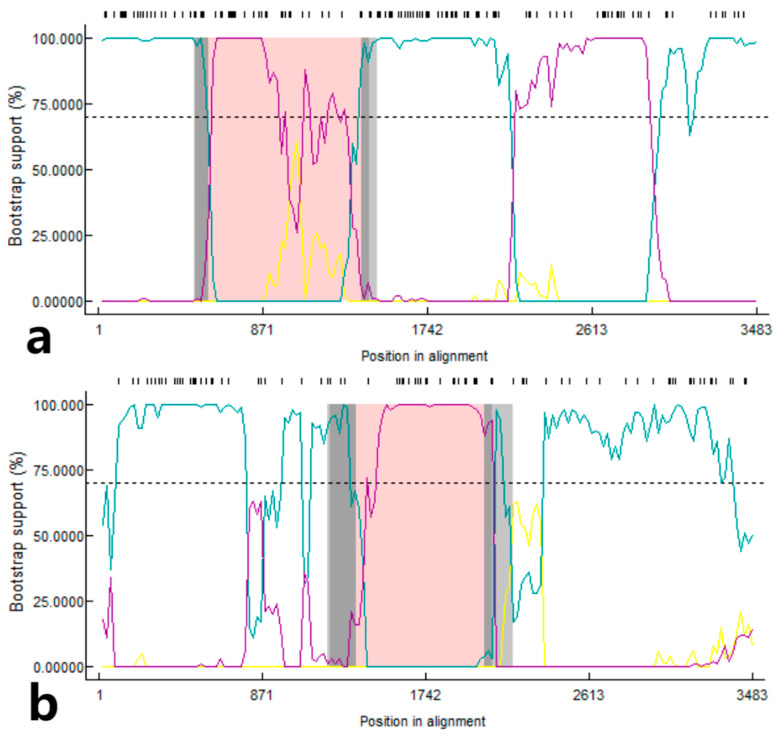
Potential recombination event analysis of the CHGX-MT505459-2019 strain (Southeast Asia lineage) and the CHGX-MT505449-2017 strain (China 1.1 lineage) based on the RDP4 software. (**a**) The PDCoV S gene of strain CHGX-MT505459-2019 recombination event analysis of potential recombination events between the CHGX-MT505446-2017 strain (Southeast Asia lineage) and the CHGX-MT505452-2018 strain (null). (**b**) The PDCoV S gene of strain CHGX-MT505449-2017 recombination event analysis of the potential recombination events between CHGX-MN173782-2016 strain (China 1.1 lineage) and the CHGX-MT505447-2017 strain (China 1.1 lineage).

**Figure 7 microorganisms-12-00416-f007:**
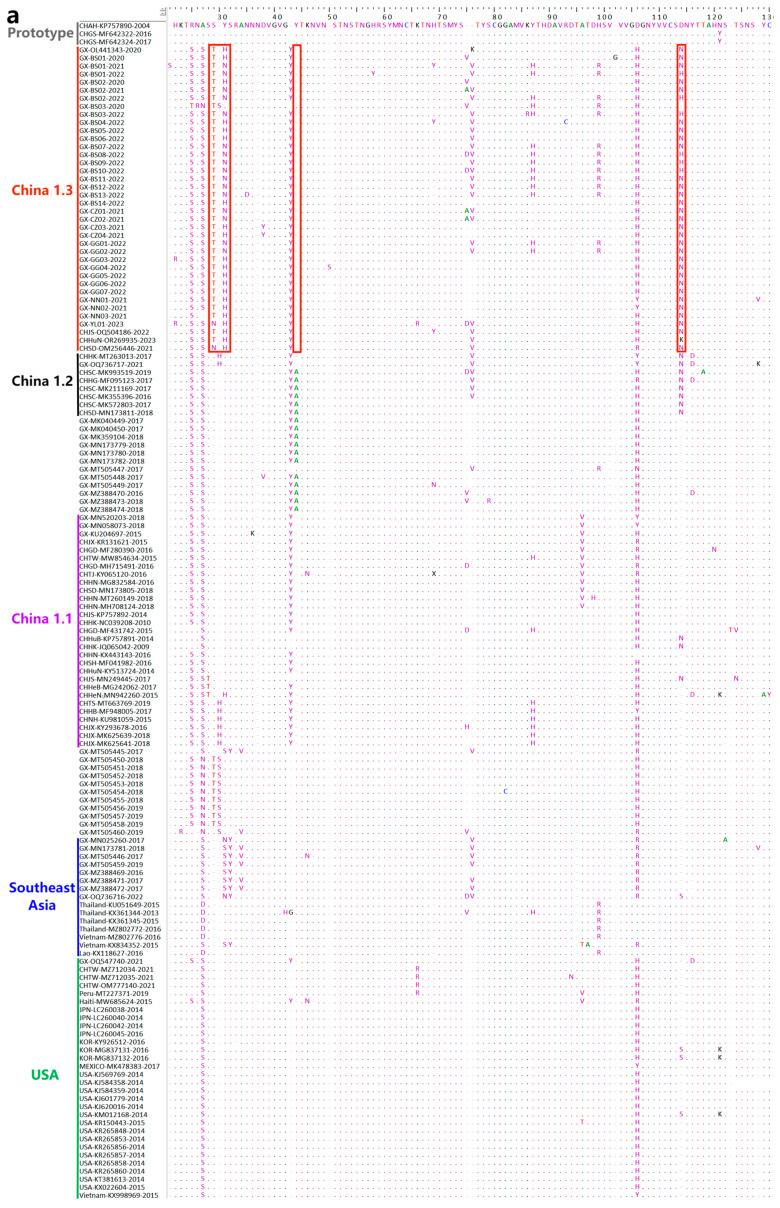
The amino acid comparison of PDCoV S gene. The images have been labeled according to the branches of the genetic evolutionary tree, and the red boxes show the region where the China 1.3 lineage is different from the other branches. (**a**–**d**) represent the order of the sequence diagram.

**Table 1 microorganisms-12-00416-t001:** Primers for amplification of the PDCoV S, M, and N genes.

Gene	Primer	Sequence (5′→3′)	Product/bp
S	PDCoV-S1-F PDCoV-S1-R PDCoV-S2-F PDCoV-S2-R PDCoV-S3-F PDCoV-S3-R PDCoV-S4-F PDCoV-S4-R PDCoV-S5-F PDCoV-S5-R	CTCGGCTCGTGAGTTAGAGAAG GACCCCGATACAACCTAACATAG CTTCCACCTGATTTAACTGAC GGGTTTTGCCAGTGGTTATA ATGGCATAGCATTCACCTCTC TGCCATTTTCTCAGCATCAAC AATGGCATCATGGTTCTAC CGCTGCAATCAGTTAATTG CGGCAAGCTGATTTCATAC CGCCAAGCGCCCATATGATC	763 865 824 853 823
M	PDCoV- M -F PDCoV- M -R	AACCCCGTACCTGAGGATGAG GCCGCTTATTGCACAATCTA	766
N	PDCoV-N1-F PDCoV-N1-R PDCoV-N2-F PDCoV-N2-R	CCATCGCTCCAAGTCATTC CGCCTGAAAGTTGCTCTCA CAAGCTCCCAAGCGGACTTTAC CCAGCCCTAGTTTGCATGATAG	821 588

**Table 2 microorganisms-12-00416-t002:** Homology of different PDCoV branches in China.

Gene	China 1.1 Lineage	China 1.2 Lineage	China 1.3 Lineage
S	96.6–100.0%	97.9–100.0%	96.2–99.9%

**Table 3 microorganisms-12-00416-t003:** The genetic evolutionary rates of PDCoV S, M, and N genes in Guangxi province.

Gene	Mean Evolutionary Rate (Substitutions/Site/Year)	95% HPD Interval
S	1.907 × 10^−3^	1.5294 × 10^−3^–2.3232 × 10^−3^
M	8.321 × 10^−4^	6.0069 × 10^−4^–1.0886 × 10^−3^
N	1.135 × 10^−3^	7.3867 × 10^−4^–1.5594 × 10^−3^

## Data Availability

Data are contained within the article and Appendix A.

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
