# Peer review of "Genetic and Evolutionary Analysis of Porcine Deltacoronavirus in Guangxi Province, Southern China, from 2020 to 2023"

_microorganisms, 2024, doi:10.3390/microorganisms12020416_

Round 1
Reviewer 1 Report
Comments and Suggestions for Authors
This manuscript is relevant for the field since the presented results provide additional information on the prevalence and evolution of PDCoV in Guangxi province. However, I do suggest certain things, which need attention, improvement and clarification to support and strengthen the overall impact of the article.
Major points for attention:
Abstract: The abstract should be a total of about 200 words maximum, please revise.
Introduction:
I suggest citing of the latest classification released (International Committee on the Taxonomy of Viruses, ICTV.)
Methodology:
Authors should indicate the storing conditions of prepared specimens (indicate whether they were processed immediately after preparation/used for RNA extraction or stored). Also, the storing conditions of RNA isolates should be indicated (were the RNA isolates stored before qRT-PCR or used immediately after extraction).
Authors should describe RNA extraction and reverse transcription in more details. The experimental part should be written clearly and with sufficient detail about the protocol used to allow the work to be repeated. Detailed descriptions are only required for new techniques and procedures, while known methods must be cited in the references. For chemicals and apparatus used, full data should be given, including the name of the product, the company/manufacturer (do not cite a supplier, only manufacturers), the city and country (state and country) of origin.
Authors didn’t mention whether the positive and negative controls were used for qRT-PCR and Rt-PCR (if they were used, they should be described in detail).
Also, authors should describe how did they monitor the quality of RNA isolates. Was the exogenous Internal Positive Control (IPC) added to each sample to supervise the appearance of potential PCR inhibitors?
Authors should indicate the chemicals used for purification of PCR products.
Results:
Line 140: Please rephrase the sequence. 4,897 clinical specimens were tested (not detected) using multiplex RT-qPCR.
Discussion:
Line267: Please rephrase the sequence.
In Discussion results should be discussed, evaluated or interpreted not repeated (for example, Lines276-278).
Conclusions:
Please, don’t repeat the results (lines 336-344). Link the Conclusions with the goals of the study, but avoid unqualified statements and conclusions not adequately supported by the data. Conclusions should concisely and clearly explain the significance and novelty of the results obtained in the presented work.
Author Response
Revision Notes
February 11, 2023
Dear editor,
We have made all efforts to meet the comments and suggestions provided during the first review of this manuscript (Manuscript ID: 2854249), and a detailed outline of our responses to them is described in a Revision Notes.
We have prepared a Revision Notes in point-by-point fashion. In the revised manuscript, we have responded the reviewer′s comments and suggestions as specific as possible and indicated the line numbers in the revised manuscript where we have addressed each comment.
We have prepared a revised manuscript (word document), highlighting the changes we have made.
Sincerely yours,
Kaichuang Shi
Responses to reviewers
Reviewer 1:
Comments and Suggestions for Authors
This manuscript is relevant for the field since the presented results provide additional information on the prevalence and evolution of PDCoV in Guangxi province. However, I do suggest certain things, which need attention, improvement and clarification to support and strengthen the overall impact of the article.
Major points for attention:
- Abstract:The abstract should be a total of about 200 words maximum, please revise.
Response: We agree to the reviewer’s suggestions. The Abstract has been revised to less than 200 words. Please see the Abstract in the revised manuscript. Please see Lines 13-29 in the revised manuscript.
Introduction:
- I suggest citing of the latest classification released (International Committee on the Taxonomy of Viruses, ICTV.)
Response: We agree to the reviewer’s suggestion. The classification has been cited according to the latest classification released by ICTV. Please see reference 5 in the revised manuscript.
Methodology:
- Authors should indicate the storing conditions of prepared specimens (indicate whether they were processed immediately after preparation/used for RNA extraction or stored). Also, the storing conditions of RNA isolates should be indicated (were the RNA isolates stored before qRT-PCR or used immediately after extraction).
Response: We agree to the reviewer’s suggestions. The storing conditions of prepared specimens, and the storing conditions of RNA isolates have been indicated in the revised manuscript. Please see Lines 68-70, and Lines 72-75 in the revised manuscript.
- Authors should describe RNA extraction and reverse transcription in more details.
Response: We agree to the reviewer’s suggestions. RNA extraction and reverse transcription has been described in more details. Please see Lines 71-75 in the revised manuscript.
- The experimental part should be written clearly and with sufficient detail about the protocol used to allow the work to be repeated. Detailed descriptions are only required for new techniques and procedures, while known methods must be cited in the references.
Response: We agree to the reviewer’s suggestions. The techniques and procedures for detection of the clinical specimens has been described in more details, while the known methods have been cited in the references and described in briefly. Please see Lines 76-101 in the revised manuscript.
- For chemicals and apparatus used, full data should be given, including the name of the product, the company/manufacturer (do not cite a supplier, only manufacturers), the city and country (state and country) of origin.
Response: We agree to the reviewer’s suggestions. The more detailed information on the chemicals and apparatus used in this study has been added in the revised manuscript. Please see 2.1 and 2.2 in the part Materials and Methods in the revised manuscript.
- Authors didn’t mention whether the positive and negative controls were used for qRT-PCR and Rt-PCR (if they were used, they should be described in detail).
Response: We agree to the reviewer’s suggestions. The positive and negative controls were used for RT-qPCR and RT-PCR. Please see Lines 89-92 in the revised manuscript.
- Also, authors should describe how did they monitor the quality of RNA isolates. Was the exogenous Internal Positive Control (IPC) added to each sample to supervise the appearance of potential PCR inhibitors?
Response: We agree to the reviewer’s suggestions. In this study, the recombinant standard plasmid construct containing PDCoV M gene fragment was used as positive control, and the negative fecal and nuclease-free sterilized distilled water were used as negative controls. The exogenous Internal Positive Control (IPC) did not add to each sample. Please see Lines 89-92 in the revised manuscript.
- Authors should indicate the chemicals used for purification of PCR products.
Response: We agree to the reviewer’s suggestions. The chemicals used for purification of PCR products have been described in the revised manuscript. Please see Lines 121-122 in the revised manuscript.
Results:
- Line 140: Please rephrase the sequence. 4,897 clinical specimens were tested (not detected) using multiplex RT-qPCR.
Response: We agree to the reviewer’s suggestions. The word “detected” has been changed to “tested”. Please see Line 177 in the revised manuscript.
Discussion:
- Line267: Please rephrase the sequence.
Response: We agree to the reviewer’s suggestions. The sequence has been rephrased. Please see Lines 304-306.
- In Discussion results should be discussed, evaluated or interpreted not repeated (for example, Lines276-278).
Response: We agree to the reviewer’s suggestions. The Discussion has been revised carefully. The sentence mentioned has been re-written. Please see Lines 313-315.
Conclusions:
- Please, don’t repeat the results (lines 336-344). Link the Conclusions with the goals of the study, but avoid unqualified statements and conclusions not adequately supported by the data. Conclusions should concisely and clearly explain the significance and novelty of the results obtained in the presented work.
Response: We agree to the reviewer’s suggestions. The Conclusion has been revised carefully. Please see Lines 374-388 in the part Conclusion in the revised manuscript.
Reviewer 2:
Comments and Suggestions for Authors
A good effort by the author.
Minor clarifications are required which are as under:
- Sample shipment procedure from the farms to the lab.
Response: We agree to the reviewer’s suggestions. The sample shipment procedure from the farms to the lab has been described in the revised manuscript. Please see Lines 68-70 in the revised manuscript.
- Total RNA extraction kit detail including cat No and company is required.
Response: We agree to the reviewer’s suggestions. The detailed information on the total RNA extraction kit has been described in the revised manuscript. Please see Lines 72-75 in the revised manuscript.
- RT-PCR kit details including cat No and company is required.
Response: We agree to the reviewer’s suggestions. The detailed information of the RT-PCR kit has been described in the revised manuscript. Please see Lines 116-117 in the revised manuscript.
- A very brief SOP of the multiplex RTPCR would have been a good addition for reproducibility in other labs if required.
Response: We agree to the reviewer’s suggestions. The brief procedure of the multiplex RT-qPCR has been described in Lines 76-101 in the revised manuscript, and the brief procedure of the RT-PCR has been described in Lines 113-120 in the revised manuscript.
- This protocol must be explained in detail with primer/probe details etc.
Response: We agree to the reviewer’s suggestions. The detailed information on primer/probe has been described in the revised manuscript. Please see Lines 80-89 in the revised manuscript.
- The justification of randomly selecting 34 specimens for sequencing should be provided. In my opinion, the specimen selection should have be strategical instead of random. It should have been area wise and year wise.
Response: We agree to the reviewer’s suggestions. The 34 positive specimens were selected for gene sequencing basing on different regions, and different pig farms in Guangxi province, different seasons of the year, and having Ct values less than 25 cycles. Please see Lines 109-111.
- Please provide software versions used. Moreover, please mention if the software used for recombination analysis RDP4 is either a public website or private?
Response: The Recombination Detection Program (RDP4) software (http://www.bioinf.manchester.ac.uk/recombination/programs) was downloaded from public website. Please see Lines 166-169 in the revised manuscript.
Comments on the Quality of English Language
- The quality of English language is good. Please make one correction at Line 140 in which you wrote 4897 specimens were "detected". Suggested to change it to "tested" please.
Response: We agree to the reviewer’s suggestions. The word “detected” has been changed to “tested”. Please see Line 177 in the revised manuscript.

Reviewer 2 Report
Comments and Suggestions for Authors
A good effort by the author.
Minor clarifications are required which are as under:
1. Sample shipment procedure from the farms to the lab.
2. Total RNA extraction kit detail including cat No and company is required.
3. RT-PCR kit details including cat No and company is required.
4. A very brief SOP of the multiplex RTPCR would have been a good addition for reproducibility in other labs if required.
5. This protocol must be explained in detail with primer/probe details etc.
6. The justification of randomly selecting 34 specimens for sequencing should be provided. In my opinion, the specimen selection should have be strategical instead of random. It should have been area wise and year wise.
7. Please provide software versions used. Moreover, please mention if the software used for recombination analysis RDP4 is either a public website or private?
Thank you.
Comments on the Quality of English LanguageThe quality of English language is good. Please make one correction at Line 140 in which you wrote 4897 specimens were "detected". Suggested to change it to "tested" please.
Author Response

(The authors gave the same response as above.)
